# Second victim syndrome in intensive care unit healthcare workers: A systematic review and meta-analysis on types, prevalence, risk factors, and recovery time

**Kazuaki Naya** [1], **Gen Aikawa** [2], **Akira Ouchi** [2], **Mitsuki Ikeda** [3], **Ayako Fukushima** [4], **Shuhei Yamada** [1], **Megumi Kamogawa** [4], **Shun Yoshihara** [4], **Hideaki Sakuramoto** [4] *

1 Department of Adult Health Nursing, Tokyo Healthcare University Wakayama Faculty of Nursing, Wakayama City, Wakayama, Japan, 2 Department of Adult Health Nursing, College of Nursing, Ibaraki Christian University, Hitachi, Ibaraki, Japan, 3 Department of Emergency and Critical Care Medicine, Graduate School of Comprehensive Human Sciences, University of Tsukuba, Tsukuba City, Ibaraki, Japan, 4 Department of Critical Care and Disaster Nursing, Japanese Red Cross Kyushu International College of Nursing, Munakata, Fukuoka, Japan

☉ These authors contributed equally to this work.
* gongehead@yahoo.co.jp

**Data Availability Statement:** All relevant data are within the paper and its Supporting information files.

## Abstract

### Introduction

Patient safety incidents, including medical errors and adverse events, frequently occur in intensive care units, leading to a significant psychological burden on healthcare workers. This burden results in second victim syndrome, which impacts the psychological and psychosomatic well-being of these workers. However, a systematic review focusing specifically on this condition among intensive care unit healthcare workers is lacking. Therefore, we aimed to conduct a systematic review and meta-analysis to examine the occurrence of second victim syndrome among intensive care unit healthcare workers, including the types, prevalence, risk factors, and recovery time associated with this condition.

### Methods

We conducted a comprehensive search of the MEDLINE, CINAHL, PsycINFO, and Igaku Chuo Zasshi databases. The eligibility criteria encompassed retrospective, prospective, and cross-sectional studies and controlled trials, with no language restrictions. Data on the type, prevalence, risk factors, and recovery time of second victim syndrome were extracted and pooled. Prevalence estimates from the included studies were combined using a random-effects meta-analytic model.

### Results

Of the 2,245 records retrieved, 16 potentially relevant studies were identified. Following full-text evaluation, five studies met the inclusion criteria and were included in the review. The findings revealed that 58% of intensive care unit healthcare workers experienced second

**Funding:** AF, Japanese Red Cross Kyushu International College of Nursing (22-1) GA, TERUMO LIFE SCIENCE FOUNDATION (22-□ 6001), https://www.terumozaidan.or.jp/ The funders had no role in study design, data collection and analysis, decision to publish, or preparation of the manuscript.

**Competing interests:** The authors have declared that no competing interests exist.

victim syndrome. Frequent symptoms included guilt (12–68%), anxiety (38–63%), anger at self (25–58%), and lower self-confidence (7–58%). However, specific risk factors exclusive to intensive care unit healthcare workers were not identified in the review. Furthermore, approximately 20% of individuals took more than a year to recover or did not recover at all from the second victim syndrome.

## Conclusions

Thus, this condition is prevalent among intensive care unit healthcare workers and may persist for extended periods, potentially exceeding a year. The risk factors for second victim syndrome in the intensive care unit setting are unclear and require further investigation.

## Introduction

Patient safety incidents, including medical errors and adverse events, frequently occur in clinical practice and pose a threat to patients' lives and well-being [1]. Individuals involved in these incidents, particularly healthcare workers, can also experience negative effects and are referred to as second victims [2, 3]. This phenomenon causes the second victim syndrome (SVS), characterized by psychological reactions such as anxiety and depression, as well as psychosomatic symptoms including headaches and sleep disturbances [2, 4].

Intensive care units (ICUs) are particularly susceptible to patient safety incidents, which can place a significant psychological burden on healthcare workers and lead to a high prevalence or exacerbation of SVS. In critically ill patients in the ICU, treatment is complex and high-risk, and the incidence of medical errors and adverse events is high, affecting patient outcomes [5–7]. Notably, studies have reported an 18% prevalence of SVS among ICU healthcare workers, highlighting its significance [8].

Despite the importance of understanding SVS in ICU healthcare workers, no systematic review has addressed this topic, to the best of our knowledge. Therefore, in this study, we aim to conduct a systematic review to explore SVS in ICU healthcare workers, including its types, prevalence, risk factors, and recovery time.

## Materials and methods

We conducted this systematic review and meta-analysis following the recommendations of the Joanna Briggs Institute Reviewer's Manual [9] and the Preferred Reporting Items for Systematic Reviews and Meta-analyses Statement [10]. The systematic review protocol was registered with the International Prospective Register of Systematic Reviews (registration number: CRD42023389943).

### Eligibility criteria

In this study, SVS was defined as "negative impacts on health care workers, directly or indirectly involved in an unanticipated adverse patient event, unintentional healthcare error, or patient injury" [3]. We employed the context, condition, and population framework to guide the selection of eligible studies. The inclusion criteria were as follows: (1) context—healthcare workers exposed to patient safety incidents, including harmful incidents, near misses, and no-harm incidents, as defined by the Canadian Patient Safety Institute; (2) condition—second victim symptoms such as psychological responses and psychosomatic symptoms; (3) population

—all ICU healthcare workers; and (4) study design—retrospective, prospective, and cross-sectional studies and controlled trials. No language restrictions were imposed, and studies for which full-text articles could not be obtained were excluded.

## Information sources and search strategy

We conducted searches in the MEDLINE (via PubMed), CINAHL (via EBSCOhost), PsycINFO (via Ovid), and Igaku Chuo Zasshi databases from inception to January 17, 2023. Ongoing trials were searched in the World Health Organization International Clinical Trials Registry Platform. The search terms used in all databases included "critical care," "intensive care unit," and "critical illness," cross-referenced with the terms "medical errors," "patient safety," and "second victim syndrome" In addition, relevant studies were identified through hand searches of reference lists of the identified studies and articles (based on Google Scholar) that cited those studies from the period after the second screening to March 1, 2023. No language restrictions were applied. The complete search strategy used in all databases is presented in S1 Text.

## Selection process

Of the nine reviewers (KN, GA, AO, MI, AF, SY, MK, SY, and HS), two independently reviewed the titles and abstracts to identify potentially relevant studies. Two reviewers independently assessed eligibility based on full-text reviews. Disagreements between reviewers were resolved through consensus discussion or arbitration by a third reviewer.

## Data collection and outcome process

Data extraction included the following: author name, year of publication, study design, country, sample size, and characteristics of healthcare workers. Outcome data were extracted on the types, prevalence, and risk factors of SVS and recovery time from SVS. Data were documented using an Excel spreadsheet, and data extraction was performed independently by five reviewers (KN, GA, AO, AF, and HS). Authors of studies that did not provide details about the characteristics and outcomes of ICU healthcare workers were contacted to obtain data. To pool the results, data were extracted dichotomously from individual studies. The results were pooled by category.

## Assessment of study quality

The Mixed Methods Assessment Tool (MMAT) was used to evaluate the methodological quality of each study [11]. Studies were rated on a categorical scale as "no," "can't tell," or "yes" to indicate whether they met the methodological quality criteria assessed. The number of items rated "yes" was counted to provide an overall score out of a possible 5, with a higher number corresponding to stronger methodological quality. Studies were appraised against the MMAT screening criteria by two researchers (KN and SY) independently. Instead of generating an overall score for each study, a qualitative approach was applied by providing a detailed review of study quality [11]. Studies were appraised as having a low, moderate, or high methodological quality.

## Statistical analysis

We pooled prevalence estimates from the included studies using a random-effects meta-analytic model. Heterogeneity across studies was assessed using $I^2$ statistics. $I^2$ was categorized as low (0–40%), moderate (30–60%), substantial (50–90%), or considerable (75–100%). The

results are presented as forest plots with 95% confidence intervals (CIs). The analysis was conducted using R statistical software version 4.3.0 (R Development Core Team, 2008) and the "*meta*" package.

## Ethics

Ethical approval and patient consent were not required for this study.

## Results

### Selection and inclusion of studies

The electronic databases and hand search yielded 2,245 records (2,242 from electronic databases and 3 from hand search). After screening titles and/or abstracts, 16 full-text articles were assessed for eligibility. Eleven studies were subsequently excluded for various reasons such as being unrelated to the concept or context. Ultimately, five studies that met the inclusion criteria were included in the review (Fig 1).

The included articles were predominantly written in English and published between 2002 and 2023, with one study being in Portuguese [8]. One study was conducted in Brazil [8], another in the United States [12], and the remaining three in Germany [13–15]. The three German studies were conducted by the same research team.

All studies utilized a cross-sectional survey design. While some authors calculated only descriptive statistics, others additionally applied logistic regression models or chi-squared tests. All studies employed self-report questionnaires administered via paper-and-pencil or web-based/electronic platforms. Four studies were conducted with physicians or nurses [8, 13–15]. One study was conducted with a wide range of professionals, including physicians, nurses, respiratory therapists, and pharmacists from the Society of Critical Care Medicine [12]. The survey employed medical error scenarios to investigate the symptoms experienced by healthcare providers when faced with such situations (Table 1).

The three studies conducted in Germany utilized the SeViD (Second Victims in Deutschland) questionnaire, which had been previously developed and validated by researchers [13–15]. These surveys were conducted among physicians and nurses of the German Society of Internal Medicine and the German Prehospital Emergency Physician Association.

### Quality assessment

Among the included studies, one study met 80% of the quality criteria, three met 60% of the quality criteria, and one met only 20% of the quality criteria. However, in most studies, the data were not considered representative of the general population because of sampling limitations in terms of specific regions or populations and low response rates (Table 2).

### Type and prevalence of second victim symptoms

The type and prevalence of each symptom experienced by the second victim are listed in Table 3. Symptoms were broadly categorized into thoughts, psychological distress, physical distress, impact on behavior, and impact on sociability. Frequent symptoms included guilt (12–68%), anxiety (38–63%), anger at self (25–58%), and lower self-confidence (7–57%).

### Risk factors of SVS and recovery time from SVS

In a survey conducted among clinician members of the Society of Critical Care Medicine [12], surgeons and anesthesiologists exhibited higher negative responses following procedural errors, whereas internal medicine and emergency medicine practitioners showed higher

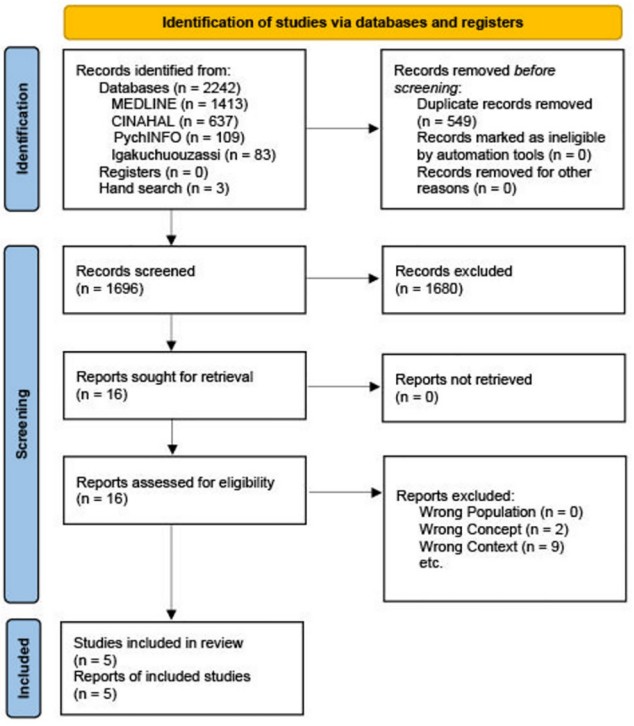

**Fig 1. PRISMA 2020 flow diagram.**

negative responses after diagnostic errors. Surveys conducted using the SeViD questionnaire did not exclusively identify risk factors for ICU healthcare workers.

The recovery time from SVS varied among ICU healthcare workers, with 2–4% recovering in less than one day, 22–29% within one week, 20–40% within one month, 10–20% within one year, 1–11% after more than one year, and 8–15% never recovering (Table 1).

## Meta-analysis of SVS prevalence

Estimates of lifetime and 12-month prevalence of SVS were calculated in three studies [13–15]. The meta-analysis showed a lifetime prevalence of SVS at 58% (95% CI = 52–63, $I^2$ = 47%) and a 12-month prevalence of SVS at 31% (95% CI = 17–49, $I^2$ = 95%) (Figs 2 and 3). High heterogeneity was observed for the 12-month prevalence. However, subgroup- and nursing-specific analyses could not be conducted because of the limited number of articles.

One study was excluded from the meta-analysis because of its use of medical error scenarios and different methods compared to other studies [12]. Another study was excluded from the analysis because it did not calculate the incidence of SVS [8].

## Discussion

Our study demonstrated that the majority of ICU healthcare workers experienced SVS, most lasting for more than one year. The risk factors for SVS in ICU healthcare workers remain unclear.

To the best of our knowledge, this is the first systematic review to focus on SVS among ICU healthcare workers. Our study showed that 58% of ICU healthcare workers had SVS, which is higher than the incidence (10–43%) reported for healthcare workers in various healthcare

**Table 1. Summarized findings of included articles.**

| Author | Country | Study design | Participants | Second victim symptoms | | Recovery time | |
|---|---|---|---|---|---|---|---|
| Padilha et al., 2002 [8] | Brazil | Cross-sectional, questionnaire survey | 148 ICU nurses from 7 hospitals | Anxiety | 38% | None | |
| | | | | Powerlessness | 15% | | |
| | | | | Guilt | 12% | | |
| | | | | Anger | 10% | | |
| | | | | Worry | 8% | | |
| | | | | Insecurity | 7% | | |
| | | | | Despair | 3% | | |
| | | | | Others | 5% | | |
| Kaur et al., 2019 [12] | United States | Cross-sectional, web-based questionnaire survey using medical error scenarios | 148 clinicians working in ICUs who were members of the Society of Critical Care Medicine | Guilt | 68% | None | |
| | | | | Anxiety | 63% | | |
| | | | | Anger at self | 58% | | |
| | | | | Professional self-doubt | 51% | | |
| | | | | Re-living the event over and over | 48% | | |
| | | | | Fear of litigation | 47% | | |
| | | | | Fear of judgment by colleagues | 46% | | |
| | | | | Loss of sleep | 44% | | |
| | | | | Shame | 27% | | |
| | | | | Defensiveness | 25% | | |
| | | | | Loss of concentration | 24% | | |
| | | | | Depression | 23% | | |
| | | | | Loss of reputation | 18% | | |
| | | | | Anger at others | 17% | | |
| | | | | Loss of interest in daily activities | 14% | | |
| | | | | Considered career change | 11% | | |
| | | | | Use of alcohol or other substances | 5% | | |
| | | | | Thoughts of hurting oneself | 3% | | |
| | | | | Did not experience any negative feelings after being involved in an adverse event | 1% | | |
| Strametz et al. [SeViD-I], 2021 [13] | Germany | Cross-sectional, web-based questionnaire survey | 128 young (≤ 35 years) physicians working in ICUs and IMCs who were members of the German Society of Internal Medicine | Fear of social isolation from colleagues | 30% | Less than one day | 4% |
| | | | | Fear of losing the job | 19% | Within one week | 29% |
| | | | | Lethargy | 34% | Within one month | 40% |
| | | | | Depressed mood | 45% | Within one year | 10% |
| | | | | Concentration problems | 36% | More than one year | 1% |
| | | | | Recall of the situation outside the workplace | 45% | Never | 10% |
| | | | | Recall of the situation at the workplace | 45% | | |
| | | | | Aggressive, risky behavior | 9% | | |
| | | | | Defensive, overprotective behavior | 44% | | |
| | | | | Psychosomatic reactions (headaches, back pain) | 28% | | |
| | | | | Difficulties sleeping or excessive need to sleep | 44% | | |
| | | | | Use of substances (alcohol/drugs) due to this event | 22% | | |
| | | | | Feeling of shame | 35% | | |
| | | | | Feeling of guilt | 48% | | |
| | | | | Lower self-confidence | 57% | | |
| | | | | Social isolation | 18% | | |
| | | | | Anger against others | 30% | | |
| | | | | Anger against oneself | 35% | | |
| | | | | Desire to get support from others | 52% | | |
| | | | | Desire to work through the incident for a deeper understanding | 52% | | |

*(Continued)*

**Table 1.** (Continued)

| Author | Country | Study design | Participants | Second victim symptoms | | Recovery time | |
|--------|---------|--------------|--------------|------------------------|---|---------------|---|
| Strametz et al. [SeViD-II], 2021 [14] | Germany | Cross-sectional, web-based questionnaire survey | 107 nurses working in ICUs and IMCs who were members of the German Society of Internal Medicine | Fear of social isolation from colleagues | 17% | Less than one day | 3% |
| | | | | Fear of losing the job | 16% | Within one week | 29% |
| | | | | Lethargy | 30% | Within one month | 20% |
| | | | | Depressed mood | 34% | Within one year | 19% |
| | | | | Concentration problems | 36% | More than one year | 2% |
| | | | | Recall of the situation outside the workplace | 25% | Never | 15% |
| | | | | Recall of the situation at the workplace | 36% | | |
| | | | | Aggressive, risky behavior | 9% | | |
| | | | | Defensive, overprotective behavior | 36% | | |
| | | | | Psychosomatic reactions (headaches, back pain) | 37% | | |
| | | | | Difficulties sleeping or excessive need to sleep | 39% | | |
| | | | | Use of substances (alcohol/drugs) due to this event | 13% | | |
| | | | | Feeling of shame | 33% | | |
| | | | | Feeling of guilt | 33% | | |
| | | | | Lower self-confidence | 39% | | |
| | | | | Social isolation | 12% | | |
| | | | | Anger against others | 27% | | |
| | | | | Anger against oneself | 25% | | |
| | | | | Desire to get support from others | 37% | | |
| | | | | Desire to work through the incident for a deeper understanding | 36% | | |
| Marung et al. [SeViD-III], 2023 [15] | Germany | Cross-sectional, web-based questionnaire survey | 184 physicians working at ICUs and IMCs who were members of the German Prehospital Emergency Physician Association | Fear of social isolation from colleagues | 22% | Less than one day | 2% |
| | | | | Fear of losing the job | 10% | Within one week | 22% |
| | | | | Lethargy | 33% | Within one month | 37% |
| | | | | Depressed mood | 38% | Within one year | 20% |
| | | | | Concentration problems | 31% | More than one year | 11% |
| | | | | Recall of the situation outside the workplace | 36% | Never | 8% |
| | | | | Recall of the situation at the workplace | 38% | | |
| | | | | Aggressive, risky behavior | 10% | | |
| | | | | Defensive, overprotective behavior | 35% | | |
| | | | | Psychosomatic reactions (headaches, back pain) | 18% | | |
| | | | | Difficulties sleeping or excessive need to sleep | 35% | | |
| | | | | Use of substances (alcohol/drugs) due to this event | 11% | | |
| | | | | Feeling of shame | 27% | | |
| | | | | Feeling of guilt | 40% | | |
| | | | | Lower self-confidence | 44% | | |
| | | | | Social isolation | 11% | | |
| | | | | Anger against others | 21% | | |
| | | | | Anger against oneself | 28% | | |
| | | | | Desire to get support from others | 36% | | |
| | | | | Desire to work through the incident for a deeper understanding | 42% | | |

ICU, intensive care unit; IMC, intermediate care unit; SeViD, Second Victims in Deutschland

**Table 2. Quality assessment of included studies using MMAT.**

| Types of mixed methods study components | Methodological quality criteria | Padilha et al. 2002 [8] | Kaur et al. 2019 [12] | Strametz et al. 2021 [13] | Strametz et al. 2021 [14] | Marung et al. 2023 [15] |
|---|---|---|---|---|---|---|
| 4. Quantitative descriptive | 4.1. Is the sampling strategy relevant to address the research question? | N | Y | Y | Y | Y |
| | 4.2. Is the sample representative of the target population? | N | Y | N | N | N |
| | 4.3. Are the measurements appropriate? | U | Y | Y | Y | Y |
| | 4.4. Is the risk of nonresponse bias low? | Y | N | N | N | N |
| | 4.5. Is the statistical analysis appropriate to answer the research question? | N | Y | Y | Y | Y |
| | Score (%) | 20 | 80 | 60 | 60 | 60 |

MMAT = Mixed Methods Appraisal Tool; Y = yes; N = no; U = cannot tell.

All studies that went through quality assessment passed the screening questions: 1) Are there clear research questions? 2) Do the collected data allow for addressing the research questions?

settings (e.g., operating rooms, obstetrics, and internal medicine) [16]. This could be because ICU healthcare workers frequently encounter patient safety incidents, and their psychological impact is significant. The ICU targets critically ill patients and provides highly invasive treatment using many complex medical devices. The frequency of medical procedures is also high, which inevitably leads to patient safety incidents such as adverse events and medical errors. Between 20% and 51% of critically ill patients experience an adverse event during their ICU stay, which is higher than an incidence of 10% in healthcare overall [6, 17, 18]. The incidence of preventable adverse events is also 18% in ICUs, higher than that in general wards [19]. These adverse events are associated with longer ICU and hospital stays [7]. Furthermore, the incidence of human error in the ICU is 31%, leading to fatal or permanent injury or prolonging the length of ICU stay [5]. Therefore, more ICU healthcare workers may be involved in patient safety incidents, and the impact of these incidents on patients may be a major psychological burden for healthcare workers.

Our findings showed that the symptoms experienced by second victims in the ICU, such as guilt, anxiety, lower self-confidence, and re-living the event repeatedly, were similar to those reported in other systematic reviews on SVS for healthcare workers [16]. A previous meta-analysis of all healthcare workers showed that "troubling memories" (81%, 95% CI = 46–95), "anxiety/concern" (76%, 95% CI = 33–95), and "anger toward themselves" (75%, 95% CI = 59–86) were the most frequent symptoms among second victims [20]. Both ICU healthcare workers and healthcare workers in general tend to have distressing memories associated with patient safety events. Unwanted, upsetting memories and flashbacks are common after traumatic experiences [21, 22]. Therefore, focusing on the presence or absence of these symptoms would be necessary.

Approximately 60% of individuals who experienced SVS recovered within one month, while approximately 20% took more than a year to recover or did not recover at all. The impact of prolonged SVS is unknown owing to lack of evidence. However, as SVS is associated with turnover intention and absenteeism, addressing patient safety events early on is necessary for healthcare providers [23]. For example, support programs such as peer support and learning from adverse events are desired by second victims, and their availability and effectiveness have been reported [24–26].

Despite the significant findings, we acknowledge certain limitations of our study. First, the meta-analysis of 12-month SVS prevalence showed high heterogeneity, which may be attributable to the variability in the probability of encountering a patient safety event within one year.

**Table 3. Prevalence of second victim symptoms.**

| Second victim symptoms | Prevalence | Reference |
|---|---|---|
| **Thought** | | |
| Guilt | 12–68% | Padilha et al. [8], Kaur et al. [12], Strametz et al. [13], Strametz et al. [14], Marung et al. [15] |
| Lower self-confidence | 7–57% | Padilha et al. [8], Kaur et al. [12], Strametz et al. [13], Strametz et al. [14], Marung et al. [15] |
| Shame | 27–35% | Kaur et al. [12], Strametz et al. [13], Strametz et al. [14], Marung et al. [15] |
| Worry | 8% | Padilha et al. [8] |
| Despair | 3% | Padilha et al. [8] |
| Fear | | |
| of judgment by colleagues | 17–46% | Kaur et al. [12], Strametz et al. [13], Strametz et al. [14], Marung et al. [15] |
| of losing the job | 10–19% | Strametz et al. [13], Strametz et al. [14], Marung et al. [15] |
| of litigation | 47% | Kaur et al. [12] |
| Anger | 10% | Padilha et al. [8] |
| at self | 25–58% | Kaur et al. [12], Strametz et al. [13], Strametz et al. [14], Marung et al. [15] |
| at others | 17–30% | Kaur et al. [12], Strametz et al. [13], Strametz et al. [14], Marung et al. [15] |
| Thoughts of hurting oneself | 3% | Kaur et al. [12] |
| Desire to work through the incident for a deeper understanding | 36–52% | Strametz et al. [13], Strametz et al. [14], Marung et al. [15] |
| Desire to get support from others | 36–52% | Strametz et al. [13], Strametz et al. [14], Marung et al. [15] |
| Considered career change | 11% | Kaur et al. [12] |
| **Psychological distress** | | |
| Anxiety | 38–63% | Padilha et al. [8], Kaur et al. [12] |
| Depression | 23–45% | Kaur et al. [12], Strametz et al. [13], Strametz et al. [14], Marung et al. [15] |
| Lethargy | 15–34% | Padilha et al. [8], Strametz et al. [13], Strametz et al. [14], Marung et al. [15] |
| Re-living the event over and over | 48% | Kaur et al. [12] |
| Recall of the situation at the workplace | 36–45% | Strametz et al. [13], Strametz et al. [14], Marung et al. [15] |
| Recall of the situation outside the workplace | 25–45% | Strametz et al. [13], Strametz et al. [14], Marung et al. [15] |
| **Physical distress** | | |
| Loss of sleep | 35–44% | Kaur et al. [12], Strametz et al. [13], Strametz et al. [14], Marung et al. [15] |
| Psychosomatic reactions (headaches, back pain) | 18–37% | Strametz et al. [13], Strametz et al. [14], Marung et al. [15] |
| **Impacts on behavior** | | |
| Concentration problems | 24–36% | Kaur et al. [12], Strametz et al. [13], Strametz et al. [14], Marung et al. [15] |
| Defensive, overprotective behavior | 25–44% | Kaur et al. [12], Strametz et al. [13], Strametz et al. [14], Marung et al. [15] |
| Aggressive, risky behavior | 9–10% | Strametz et al. [13], Strametz et al. [14], Marung et al. [15] |
| Use of alcohol or other substances | 5–22% | Kaur et al. [12], Strametz et al. [13], Strametz et al. [14], Marung et al. [15] |
| Loss of interest in daily activities | 14% | Kaur et al. [12] |

*(Continued)*

**Table 3.** (Continued)

| Second victim symptoms | Prevalence | Reference |
|---|---|---|
| **Impacts on sociability** | | |
| Loss of reputation | 18% | Kaur et al. [12] |
| Social isolation | 11–18% | Strametz et al. [13], Strametz et al. [14], Marung et al. [15] |

The heterogeneity of SVS prevalence over a lifetime is lower than this, implying that the differences in prevalence by occupation may not be large. Second, the generalizability of the findings is limited because of the scarcity of studies on SVS in ICUs and the analysis being conducted on data from a single country. This is because mental health symptoms such as anxiety and depression differ between countries and regions [27]. Furthermore, these symptoms can be influenced by the organization's patient safety culture and the health of the work environment [28]. Third, the three SeViD studies integrated in the meta-analysis were categorized as involving healthcare professionals affiliated with ICUs or intermediate care units; thus, the prevalence of SVS may not strictly involve healthcare workers only in ICUs [13–15]. In addition, the studies included in this research were mainly limited to physicians or nurses; therapists, pharmacists, engineers, and other professionals working in ICUs were not studied, and the prevalence of SVS among them is unknown.

The high prevalence of SVS among ICU healthcare workers and its potential for prolonged impact necessitate follow-up and support. However, research on SVS in ICUs is limited, and the prevalence by professional category, risk factors, prolonged impact, and desired support remain unclear. Therefore, future research investigating the prevalence of SVS by professional category, risk factors, and its prolonged impact as well as identifying the desired support resources in ICUs can provide basic data for preventing and reducing SVS in ICUs. A step-by-step approach is recommended for the desired support resources, including the help of

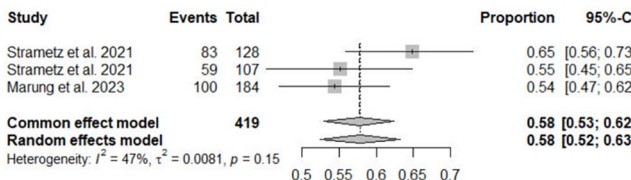

**Fig 2. Forest plot for a lifetime prevalence of second victim syndrome CI, confidence interval.**

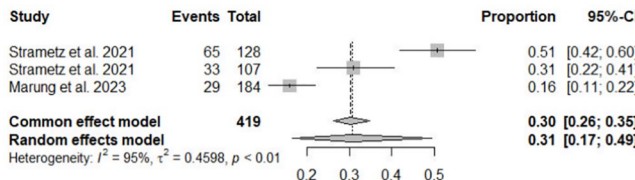

**Fig 3. Forest plot for a 12-month prevalence of second victim syndrome CI, confidence interval.**

colleagues in the department, crisis intervention by a special team, and a support network of professionals [29]. Verifying the effects of these resources in ICUs will be necessary in the future.

Further research on SVS is warranted, but caution should be exercised when investigating it. This is because it is particularly similar to concepts such as moral distress and burnout, and a distinction must be made between them. A second victim has been defined as "Any health care worker, directly or indirectly involved in an unanticipated adverse patient event, unintentional healthcare error, or patient injury, and who becomes victimized in the sense that they are also negatively impacted" by experts in a recent article [3]. Research on SVS would need to be conducted based on this definition.

## Conclusions

This systematic review highlighted that SVS is a frequently prevalent issue among ICU healthcare workers, especially with symptoms such as guilt, anxiety, lower self-confidence, and re-living the event. These symptoms can have a prolonged impact, lasting for more than a year. The findings of this study emphasized the importance of addressing SVS and providing appropriate support for ICU healthcare workers who experience patient safety incidents. Early interventions and support programs tailored to the unique needs of ICU professionals are crucial for their well-being and to mitigate the potential negative consequences of SVS. However, the risk factors for SVS in the ICU setting remain unclear and require further investigation. By gaining a deeper understanding of the factors contributing to SVS and implementing targeted interventions, healthcare organizations can create a safer and more supportive environment for their frontline staff, ultimately enhancing patient safety and well-being.

## Supporting information

**S1 Text. Search terms.**
(DOCX)

**S1 Checklist. PRISMA 2020 checklist.**
(DOCX)

## Acknowledgments

We express our appreciation to the Ibaraki Christian University Library for confirming the search formula. We also express our gratitude to the research team at SeViD for providing us with valuable data [13–15]. Additionally, we would like to acknowledge Editage (www.editage. com) for their assistance with English language editing.

## Author Contributions

**Conceptualization:** Kazuaki Naya, Gen Aikawa, Hideaki Sakuramoto.

**Data curation:** Kazuaki Naya.

**Formal analysis:** Kazuaki Naya, Gen Aikawa, Ayako Fukushima.

**Funding acquisition:** Gen Aikawa, Ayako Fukushima.

**Investigation:** Kazuaki Naya, Gen Aikawa, Akira Ouchi, Mitsuki Ikeda, Ayako Fukushima, Shuhei Yamada, Megumi Kamogawa, Shun Yoshihara, Hideaki Sakuramoto.

**Methodology:** Akira Ouchi, Hideaki Sakuramoto.

**Project administration:** Kazuaki Naya, Gen Aikawa, Hideaki Sakuramoto.

**Supervision:** Akira Ouchi, Hideaki Sakuramoto.

**Visualization:** Kazuaki Naya, Gen Aikawa, Mitsuki Ikeda, Ayako Fukushima.

**Writing – original draft:** Kazuaki Naya, Gen Aikawa, Hideaki Sakuramoto.

**Writing – review & editing:** Kazuaki Naya, Gen Aikawa, Akira Ouchi, Mitsuki Ikeda, Ayako Fukushima, Shuhei Yamada, Megumi Kamogawa, Shun Yoshihara, Hideaki Sakuramoto.

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
