## [Decision Letter · Decision Letter 0]

25 Jul 2023

PONE-D-23-18696Second victim syndrome in intensive care unit healthcare workers: A systematic review and meta-analysis on types, prevalence, risk factors, and recovery timePLOS ONE

Dear Dr. Sakuramoto,

Thank you for submitting your manuscript to PLOS ONE. After careful consideration, we feel that it has merit but does not fully meet PLOS ONE’s publication criteria as it currently stands. Therefore, we invite you to submit a revised version of the manuscript that addresses the points raised during the review process.

We look forward to receiving your revised manuscript.

Kind regards,

Amos Buh, BSc., MPH

Academic Editor

PLOS ONE

Journal Requirements:

Additional Editor Comments:

Please, carefully review this manuscript and correct any grammatical or typographical errors.

Reviewers' comments:

Reviewer's Responses to Questions

**Comments to the Author**

1. Is the manuscript technically sound, and do the data support the conclusions?

Reviewer #1: Yes

Reviewer #2: Yes

2. Has the statistical analysis been performed appropriately and rigorously? 

Reviewer #1: I Don't Know

Reviewer #2: Yes

3. Have the authors made all data underlying the findings in their manuscript fully available?

Reviewer #1: Yes

Reviewer #2: Yes

4. Is the manuscript presented in an intelligible fashion and written in standard English?

Reviewer #1: Yes

Reviewer #2: Yes

5. Review Comments to the Author

Reviewer #1: As, second victim is one of the recently emerging topic of focus, choosing the topic is one of the most appreciable thing that too in post covid era.

Few of my suggestions are as follows

1.Is there seperate focus on different types of ICU’s - as mostly MICU people tends to deal with more of death than SICU and others

2.Is there any separate categorisation of diseases,which occurs after the development of second victim syndrome, among the types of health care workers

3.Is there any requirement for treating the syndrome?

Can be highlighted

Reviewer #2: The manuscript „Second victim syndrome in intensive care unit healthcare workers: A systematic review and meta-analysis on types, prevalence, risk factors, and recovery time” is a valuable contribution to the discussion around mental health and wellbeing of health care workers in ICUs, consequences of which might directly affect patient safety in the very vulnerable group of critically patients. The manuscript is methodologically sound and is written in an intelligible fashion. However, there are a few points that we feel need to be reviewed, specified, and a little more carefully contextualized.

Authors set out to explore data on the type, prevalence, risk factors and recovery time of SVS amongst ICU health care workers. Due to the scarcity of identified studies, not all of these could be explored in depth. Have health care workers in intermediate-care units be considered too, or was the selection of eligible studies strictly limited to the intensive care setting? Is the data on recovery time all self-reported? This should be mentioned, if the case.

Second victim syndrome is a well-known psychological concept, that impairs mental health amongst the health care work force. However, SVS does not stand on its own, but is linked to other concepts such as e.g., moral distress, mattering, or burn-out. Obviously, these other syndromes are not the topic of the present manuscript but given that the manuscript also mentiones “types” of SVS as a criteria for data extraction, and one of the inclusion criteria is phrased as “second victim symptoms such as psychological responses and psychosomatic symptoms” and symptoms often overlap with the above mentioned concepts, a careful contextualization would be very desirable. See e.g., Davidson JE et al. Exploring Distress Caused by Blame for a Negative Patient Outcome. J Nurs Adm. 2016 Jan;46(1):18-24. doi: 10.1097/NNA.0000000000000288. PMID: 26575866.

Authors argue that ICU health care workers have an inherent strong self-efficacy and responsibility (is there a reference for this?), and that found high levels of “low self-confidence” caused by SVS amongst ICU health care workers might make them more “susceptible” to patient safety incidents. We would recommend a little clarification: is that meant as discussion of a feature unique to ICU workers in contrast to other health care workers? Could this have consequences with regards to proposed therapies or strategies to address the issue of SVS amongst ICU workers? Authors have included a study by Strametz et al (reference 14); those authors have published recommendations for supporting health care workers affected by SVS in the light and aftermath of the Covid-19 pandemic, which might be of interest here. Kindly refer to: Strametz R et al [Recommended actions: Reinforcing clinicians' resilience and supporting second victims during the COVID-19 pandemic to maintain capacity in the healthcare system]. Zentralbl Arbeitsmed Arbeitsschutz Ergon. 2020;70(6):264-268. German. doi: 10.1007/s40664-020-00405-7. Epub 2020 Sep 2. PMID: 32905028; PMCID: PMC7464049.

Authors mention in the discussion, that “approx. 18% of critically ill patients experienced adverse events during their ICU stay”, which is higher than in general wards. It is assumed that the reference for this statement is the cited meta-analysis by Panagioti et al. In the cited review, 18% is the pooled prevalence of preventable patient harm in the ICU, which is overlapping with, but not the same as, adverse events. While the general line of argument - namely that critically ill patients are more prone and vulnerable to mistakes, medication errors, adverse events related to poly-medication, complications of severe disease, and many other forms of adverse events - is absolutely relatable and very important, we recommend a more in-depth explanation and careful contextualization of different terminology, as well as a few more references for this important point of the discussion.

6. PLOS authors have the option to publish the peer review history of their article (what does this mean?). If published, this will include your full peer review and any attached files.

Reviewer #1: No

Reviewer #2: No

---

## [Author Response · Author response to Decision Letter 0]

27 Aug 2023

We appreciate the time and effort you and each of the reviewers have dedicated to providing insightful feedback on ways to strengthen our paper. Thus, it is with great pleasure that we resubmit our manuscript for further consideration. We have incorporated changes that reflect the detailed suggestions you have graciously provided. We also hope that our edits and the responses provided below satisfactorily address all the issues and concerns you and the reviewers have noted.

To facilitate your review of our revisions, the following are point-by-point responses to the questions and comments in your letter dated July 26, 2023.

Reviewer #1: 

As, second victim is one of the recently emerging topic of focus, choosing the topic is one of the most appreciable thing that too in post covid era.

Few of my suggestions are as follows

1.Is there seperate focus on different types of ICU’s - as mostly MICU people tends to deal with more of death than SICU and others

Thank you for your valuable comment.

We were targeting an ICU that encompasses the SICU as well as the MICU.

Therefore, in constructing the search formula, we used the term “intensive care units” without distinction.

2.Is there any separate categorisation of diseases,which occurs after the development of second victim syndrome, among the types of health care workers

Thank you for your valuable comment. There were no studies describing the diseases, which occurs after the development of second victim syndrome. Based on the SVS definition, PTSD is included in the SVS concept. We added this to the method because the SVS definition was not clearly stated:

Page 4, lines 79-81: In this study, SVS was defined as having “negative impacts on health care workers, directly or indirectly involved in an unanticipated adverse patient event, unintentional healthcare error, or patient injury” [3].

Moreover, the primary studies were divided into two categories, one with physicians and the other with nurses. The classification of each symptom by physicians and nurses is summarized in Table 1. Although the study by Kaur et al. included pharmacists and respiratory therapists, it was a scenario-based study, which can introduce considerable bias in integration. Therefore, we had no choice but to summarize the studies qualitatively.

The following sentences were added to the results section for clarity:

Page 8, lines 155-159: “Four studies were conducted with physicians or nurses [8, 13-15]. One study was conducted with a wide range of professionals, including physicians, nurses, respiratory therapists, and pharmacists from the Society of Critical Care Medicine [12]. The survey employed medical error scenarios to investigate the symptoms experienced by healthcare providers when faced with such situations (Table 1)”

Moreover, as there were no studies on ICU occupations other than those conducted with physicians and nurses, this part was added to the Limitations and Research Implications sections of the Discussion.

Page 20, lines 257-266: “In addition, the studies included in this research were mainly limited to physicians or nurses; therapists, pharmacists, engineers, and other professionals working in ICUs were not studied, and the prevalence of SVS among them is unknown.

The high prevalence of SVS among ICU healthcare workers and its potential for prolonged impact necessitate follow-up and support. However, research on SVS in ICUs is limited, and the prevalence by professional category, risk factors, prolonged impact, and desired support remain unclear. Therefore, future research investigating the prevalence of SVS by professional category, risk factors, and its prolonged impact as well as identifying the desired support resources in ICUs can provide basic data for preventing and reducing SVS in ICUs.”

3.Is there any requirement for treating the syndrome?

Can be highlighted

　

Thank you for your suggestion. The requirements for treating SVS are unclear. We could not clarify the risk factors from our results either. A recent paper stated that SVS is defined as the negative impact on healthcare professionals involved in patient safety incidents, a broader concept that does not require a medical diagnosis. Therefore, when dealing with SVS, managers and organizations need to be aware that SVS is a common condition and they should know about if their healthcare workers have symptoms. Subsequently, if the symptoms are pathological, proactive support needs to be provided. The following statement was thus added to the Research Implications section of the Discussion:

Page 20, lines 266-269: “A step-by-step approach is recommended for the desired support resources, including the help of colleagues in the department, crisis intervention by a special team, and a support network of professionals [29]. Verifying the effects of these resources in ICUs will be necessary in the future.”

Reviewer #2: 

Authors set out to explore data on the type, prevalence, risk factors and recovery time of SVS amongst ICU health care workers. Due to the scarcity of identified studies, not all of these could be explored in depth. Have health care workers in intermediate-care units be considered too, or was the selection of eligible studies strictly limited to the intensive care setting? 

　

Thank you for your valuable comment. For the German SeViD study, the background data were in the category of “ICU or IMC.” Therefore, it was not possible to perform a subgroup analysis of ICUs alone during the secondary data analysis. The following sentence was thus added to the Discussion section:

Page 19-20, lines 254-257: “Third, the three SeViD studies integrated in the meta-analysis were categorized as involving healthcare professionals affiliated with ICUs or intermediate care units; thus, the prevalence of SVS may not strictly involve healthcare workers only in ICUs [13-15].”

Is the data on recovery time all self-reported? This should be mentioned, if the case.

Thank you for your comment. It has been noted, at the beginning of the results section, that all studies were conducted in a questionnaire format and the data were self-reported.

Page 8, lines 154-155: “All studies employed self-report questionnaires administered via paper-and-pencil or web-based/electronic platforms.”

Second victim syndrome is a well-known psychological concept, that impairs mental health amongst the health care work force. However, SVS does not stand on its own, but is linked to other concepts such as e.g., moral distress, mattering, or burn-out. Obviously, these other syndromes are not the topic of the present manuscript but given that the manuscript also mentiones “types” of SVS as a criteria for data extraction, and one of the inclusion criteria is phrased as “second victim symptoms such as psychological responses and psychosomatic symptoms” and symptoms often overlap with the above mentioned concepts, a careful contextualization would be very desirable. See e.g., Davidson JE et al. Exploring Distress Caused by Blame for a Negative Patient Outcome. J Nurs Adm. 2016 Jan;46(1):18-24. doi: 10.1097/NNA.0000000000000288. PMID: 26575866.

Thank you for your comment. As you pointed out, SVS is associated with moral distress and burnout, which must be carefully considered. However, moral distress and burnout differ from SVS in that “directly or indirectly involved in an unanticipated adverse patient event, unintentional healthcare error, or patient injury.” The following is the latest definition of second victim provided by experts:

A second victim is defined as: “Any health care worker, directly or indirectly involved in an unanticipated adverse patient event, unintentional healthcare error, or patient injury, and who becomes victimized in the sense that they are also negatively impacted.”

We excluded the concepts of burnout and moral distress by including terms related to patient safety incidents in the construction of the search strategy.

We thus followed the abovementioned definition for our screening. The definition was not described in the text, so we added the following statements to the Materials and methods and Research Implications sections of the Discussion:

Page 4, lines 79-81: In this study, SVS was defined as “negative impacts on health care workers, directly or indirectly involved in an unanticipated adverse patient event, unintentional healthcare error, or patient injury” [3].

Pages 20-21, lines 270-276: “Further research on SVS is warranted, but caution should be exercised when investigating it. This is because it is particularly similar to concepts such as moral distress and burnout, and a distinction must be made between them. A second victim has been defined as “Any healthcare worker, directly or indirectly involved in an unanticipated adverse patient event, unintentional healthcare error, or patient injury, and who becomes victimized in the sense that they are also negatively impacted” by experts in a recent article [3]. Research on SVS would need to be conducted based on this definition.”

Authors argue that ICU health care workers have an inherent strong self-efficacy and responsibility (is there a reference for this?), and that found high levels of “low self-confidence” caused by SVS amongst ICU health care workers might make them more “susceptible” to patient safety incidents. We would recommend a little clarification: is that meant as discussion of a feature unique to ICU workers in contrast to other health care workers? Could this have consequences with regards to proposed therapies or strategies to address the issue of SVS amongst ICU workers? Authors have included a study by Strametz et al (reference 14); those authors have published recommendations for supporting health care workers affected by SVS in the light and aftermath of the Covid-19 pandemic, which might be of interest here. Kindly refer to: Strametz R et al [Recommended actions: Reinforcing clinicians' resilience and supporting second victims during the COVID-19 pandemic to maintain capacity in the healthcare system]. Zentralbl Arbeitsmed Arbeitsschutz Ergon. 2020;70(6):264-268. German. doi: 10.1007/s40664-020-00405-7. Epub 2020 Sep 2. PMID: 32905028; PMCID: PMC7464049.

Thank you for pointing this out. We thought that the strong sense of responsibility could easily lead to a loss of self-confidence. As you pointed out, the statement that healthcare professionals in the ICU have a strong sense of self-efficacy and responsibility was not well founded. Therefore, we have removed this paragraph.

A statement regarding support for this paper has been added to the discussion:

Page 20, lines 266-269: “A step-by-step approach is recommended for the desired support resources, including the help of colleagues in the department, crisis intervention by a special team, and a support network of professionals [29]. Verifying the effects of these resources in ICUs will be necessary in the future.”

Authors mention in the discussion, that “approx. 18% of critically ill patients experienced adverse events during their ICU stay”, which is higher than in general wards. It is assumed that the reference for this statement is the cited meta-analysis by Panagioti et al. In the cited review, 18% is the pooled prevalence of preventable patient harm in the ICU, which is overlapping with, but not the same as, adverse events. While the general line of argument - namely that critically ill patients are more prone and vulnerable to mistakes, medication errors, adverse events related to poly-medication, complications of severe disease, and many other forms of adverse events - is absolutely relatable and very important, we recommend a more in-depth explanation and careful contextualization of different terminology, as well as a few more references for this important point of the discussion.

Thank you for your valuable comment.

You are correct in that the paper we presented described preventable adverse events. As adverse events and medical errors are two different things, we decided to describe the incidence and impact of each in the ICU.

The discussion was thus revised and the following text was added:

Pages 17-18, lines 215-227: “This could be because ICU healthcare workers frequently encounter patient safety incidents, and their psychological impact is significant. The ICU targets critically ill patients and provides highly invasive treatment using many complex medical devices. The frequency of medical procedures is also high, which inevitably leads to patient safety incidents such as adverse events and medical errors. Between 20% and 51% of critically ill patients experience an adverse event during their ICU stay, which is higher than an incidence of 10% in healthcare overall [6, 17, 18]. The incidence of preventable adverse events is also 18% in ICUs, higher than that in general wards [19]. These adverse events are associated with longer ICU and hospital stays [7]. Furthermore, the incidence of human error in the ICU is 31%, leading to fatal or permanent injury or prolonging the length of ICU stay [5]. Therefore, more ICU healthcare workers may be involved in patient safety incidents, and the impact of these incidents on patients may be a major psychological burden for healthcare workers.”

Once again, thank you for giving us the opportunity to strengthen our manuscript with your valuable comments and queries. We have worked hard to incorporate your feedback and hope that these revisions persuade you to accept our submission.

---

## [Decision Letter · Decision Letter 1]

14 Sep 2023

Second victim syndrome in intensive care unit healthcare workers: A systematic review and meta-analysis on types, prevalence, risk factors, and recovery time

PONE-D-23-18696R1

Dear Dr. Sakuramoto,

We’re pleased to inform you that your manuscript has been judged scientifically suitable for publication and will be formally accepted for publication once it meets all outstanding technical requirements.

Kind regards,

Amos Buh, BSc., MPH

Academic Editor

PLOS ONE

Additional Editor Comments (optional):

Thank you for revising the manuscript and addressing reviewers comments adequately.

Reviewers' comments:

Reviewer's Responses to Questions

**Comments to the Author**

1. If the authors have adequately addressed your comments raised in a previous round of review and you feel that this manuscript is now acceptable for publication, you may indicate that here to bypass the “Comments to the Author” section, enter your conflict of interest statement in the “Confidential to Editor” section, and submit your "Accept" recommendation.

Reviewer #1: All comments have been addressed

2. Is the manuscript technically sound, and do the data support the conclusions?

Reviewer #1: Yes

3. Has the statistical analysis been performed appropriately and rigorously? 

Reviewer #1: Yes

4. Have the authors made all data underlying the findings in their manuscript fully available?

Reviewer #1: Yes

5. Is the manuscript presented in an intelligible fashion and written in standard English?

Reviewer #1: Yes

6. Review Comments to the Author

Reviewer #1: The suggestions have been well addressed with pertaining references in this revised manuscript paper.

7. PLOS authors have the option to publish the peer review history of their article (what does this mean?). If published, this will include your full peer review and any attached files.

Reviewer #1: No

---

## [Editor Report · Acceptance letter]

25 Sep 2023

PONE-D-23-18696R1 

Second victim syndrome in intensive care unit healthcare workers: A systematic review and meta-analysis on types, prevalence, risk factors, and recovery time 

Dear Dr. Sakuramoto:

I'm pleased to inform you that your manuscript has been deemed suitable for publication in PLOS ONE. Congratulations! Your manuscript is now with our production department. 

Kind regards, 

on behalf of

Mr. Amos Buh 

Academic Editor

PLOS ONE